# Contrastive Learning with Adversarial Examples

**Chih-Hui Ho**   **Nuno Vasconcelos**
Department of Electrical and Computer Engineering
University of California, San Diego
{chh279, nvasconcelos}@ucsd.edu

## Abstract

Contrastive learning (CL) is a popular technique for self-supervised learning (SSL) of visual representations. It uses pairs of augmentations of unlabeled training examples to define a classification task for pretext learning of a deep embedding. Despite extensive works in augmentation procedures, prior works do not address the selection of challenging negative pairs, as images within a sampled batch are treated independently. This paper addresses the problem, by introducing a new family of adversarial examples for constrastive learning and using these examples to define a new adversarial training algorithm for SSL, denoted as CLAE. When compared to standard CL, the use of adversarial examples creates more challenging positive pairs and adversarial training produces harder negative pairs by accounting for all images in a batch during the optimization. CLAE is compatible with many CL methods in the literature. Experiments show that it improves the performance of several existing CL baselines on multiple datasets.

## 1   Introduction

Deep networks have enabled significant advances in many machine learning tasks over the last decade. However, this usually requires supervised learning, based on large and carefully curated datasets. Self-supervised learning (SSL) [25] aims to alleviate this limitation, by leveraging unlabeled data to define surrogate tasks that can be used for network training. Early advances in SSL were mostly due to the introduction of many different surrogate tasks [2, 51, 34, 78, 81, 33], including solving image [47, 27] or video [3] puzzles, filling image patches [51, 15, 44] or discriminating between image rotations [18]. Recently, there have also been advances in learning techniques specifically tailored to SSL, such as contrastive learning (CL) [78, 12, 22, 74, 63, 23, 55, 39], which is the focus of this paper. CL is based on a surrogate task that treats instances as classes and aims to learn an invariant instance representation. This is implemented by generating a pair of examples per instance, and feeding them through an encoder, which is trained with a constrastive loss. This encourages the embeddings of pairs generated from the same instance, known as positive pairs, to be close together and embeddings originated from different instances, known as negative pairs, to be far apart.

The design of positive pairs is one of the research focuses of CL [5]. These pairs can include data from one or two modalities. For single-modality approaches, a common procedure is to rely on data augmentation techniques. For example, instances from image datasets are frequently subject to transformations such as rotation, color jittering, or scaling [12, 78, 22, 74, 39] to generate the corresponding pairs. This type of data augmentation has been shown critical for the success of CL, with different augmentation approaches having a different impact on SSL performance [12]. For video datasets, positive pairs are usually derived from temporal coherence constraints [56, 21]. For multi-view data, augmentations can be more elaborate. For example, [63] considers augmentations of color channels, depth or surface normal and shows that the performance of CL improves as the augmentations increase in diversity. Multi-modal CL approaches tend to rely on audio and video from a common video clip to design positive pairs [52]. In general, CL benefits from definitions of positive pairs that pose a greater challenge to the learning of the invariant instance representation.

Unlike the plethora of positive pair selection proposals, the design of negative pairs has received less emphasis in the CL literature. Since CL resembles approaches such as noise contrastive estimation (NCE) [20] and N-pair [59] losses, it can leverage hard negative pair mining schemes proposed for these approaches [7, 59]. However, because they treat each instance independently, previous SSL works do not consider CL algorithms that select difficult negative pairs within a batch. This is unlike CL methods based on metric learning [30, 59, 54, 62], which seek to construct batches with challenging negative samples.

In this work, we seek a general algorithm for the generation of diverse positive and challenging negative pairs. This is framed as the search for instance augmentation sets that induce the largest optimization cost for CL. A natural approach to synthesize these sets is to leverage adversarial examples [79, 10, 19, 11, 76, 76], which are crafted to attack the network and can thus be seen as the most challenging examples that it can process. We note that the goal is not to enhance robustness to adversarial attacks, but to produce a better representation for SSL. This is in line with recent studies in the adversarial literature, showing that adversarial examples can be used to improve supervised [75, 77] and semi-supervised [41] learning. We explore whether the same benefits can ensue for SSL. One difficulty is, however, that no attention has been previously devoted to the design of adversarial attacks for SSL, where no class labels are available. In fact, for SSL embeddings trained by CL, the classical definition of adversarial attack does not even apply, since CL operates on pairs of examples. We show, however, that it is possible to leverage the interpretation of CL as instance classification to produce a sensible generalization of classification attacks to the CL problem. The new attacks are then combined with recent techniques from the adversarial literature [75, 77], which treat adversarial training as multi-domain training, to produce more invariant representations for SSL.

Overall, the paper makes three contributions. First, we show that adversarial data augmentation can be used to improve the performance of SSL learning. Second, we propose a novel procedure for training *Contrastive Learning with Adversarial Examples (CLAE)* for SSL models. Unlike the attacks classically developed in the supervised learning literature, the new attacks produce pairs of examples that account for both the positive and negative pairs in a batch to maximize contrastive loss. To the best of our knowledge neither the use of attacks to improve SSL nor the design of adversarial examples with this property have been previously discussed in the literature. Finally, extensive experiments demonstrate that (1) adversarial examples can indeed be leveraged to improve CL, and (2) CLAE boosts the performance of several CL baselines across different datasets.

## 2   Related work

Since this work focuses on image classification tasks, our survey of previous work concentrates on contrastive learning (CL) and adversarial examples for image classification.

### 2.1   Contrastive learning

Contrastive learning has been widely used in the metric learning literature [13, 71, 54] and, more recently, for self-supervised learning (SSL) [68, 74, 78, 63, 22, 12, 39, 55, 23], where it is used to learn an encoder in the pretext training stage. Under the SSL setting, where no labels are available, CL algorithms aim to learn an invariant representation of each image in the training set. This is implemented by minimizing a contrastive loss evaluated on pairs of feature vectors extracted from data augmentations of the image. While most CL based SSL approaches share this core idea, multiple augmentation strategies have been proposed [74, 78, 63, 22, 12, 39]. Typically, augmentations are obtained by data transformation (i.e. rotation, cropping, random grey scale and color jittering) [78, 12], but there have also been proposals to use different color channels, depth, or surface normals as the augmentations of an image [63]. Another approach is to use an augmentation dictionary composed of the embedding vectors from the previous epoch [74] or obtained by forwarding an image through a momentum updated encoder [22]. This diversity of approaches to the synthesis of augmentations reflects the critical importance of using semantically similar example pairs in CL [5]. This has also been studied empirically in [12], showing that stronger data augmentations improve CL performance.

Despite this wealth of augmentation proposals for SSL, most CL algorithms fail to mine hard negative pairs or relate the image instances within a batch. While [70, 40, 12, 66] have mentioned the importance of selecting negative pairs, they do not propose a systematic algorithm to do this. Since

CL is inspired by the noise contrastive estimation (NCE) [20] and N-pair [59] loss methods from metric learning, it inherits the well known difficulties of hard negative mining in this literature [73, 59, 54, 62, 30, 58, 50, 60]. For metric learning methods [54, 60, 30, 59], the number of possible positive and negative pairs increases dramatically (for example, cubically when the triplet loss is used [54]) as the dataset increases. A solution used by NCE is to draw negative samples from a noise distribution that treats all negative samples equally[7, 8].

Unlike all these prior efforts, this work proposes to use "adversarial augmentations" as challenging training pairs that maximize the contrastive loss. However, unlike hard negative mining in metric learning, no class labels are provided in SSL. Hence, the consideration of how all images in the batch relate to each other is necessary for generating hard negative pairs.

## 2.2 Adversarial examples

Adversarial examples are created from clean examples to produce adversarial attacks that induce a network in error [79, 10, 19]. They have been used in many supervised learning scenarios, including image classification [19, 31, 48, 43, 9, 42, 61], object detection [11, 76, 16, 82] and segmentation [76, 4]. Typically, to defend against such attacks and increase network robustness, the network is trained with both clean and adversarial examples, a process referred as adversarial training [57, 65, 32, 35, 79, 77]. It is also possible to leverage SSL to increase robustness against unseen attacks [45]. While adversarial training is usually effective as a defense mechanism, there is frequently a decrease in the accuracy of the classification of clean examples [80, 64, 67, 75, 35].

This effect has been attributed to overfitting to the adversarial examples [35] but remains somewhat of a paradox, since the increased diversity of adversarial examples could, in principle, improve standard training [24], e.g. by enabling models trained on adversarial examples to generalize better to unseen data [69, 36]. In summary, while adversarial examples could assist learning, it remains unclear how to do this. Recently, [75, 77] have made progress along this direction, by introducing a procedure, denoted AdvProp, that treats clean and adversarial examples as samples from different domains, and uses a different set of batch normalization (BN) layers for each domain. This aims to align the statistics of the embedding of clean and adversarial samples, such that both can contribute to the network learning, and has been previously shown successful for multi-domain classification problems [6, 53].

The proposed framework CLAE is inspired by recent advances in the adversarial example literature, yet parallel to them. Unlike these methods, we aim to leverage the strength of adversarial example for SSL, where no labels are available, and the focus on pairs rather than single examples requires an altogether different definition of adversaries. Our aims is to use adversarial training to compensate the limitation of current CL algorithms, by both generating challenging positive pairs and mining effective hard negative pairs for the optimization of the contrastive loss. Note that the goal is to produce better embeddings for CL, rather than robustifying CL embeddings against attacks.

# 3 Leveraging adversarial examples for improved contrastive learning

In this section, we introduce the approach proposed to create adversarial examples for CL, and a novel training scheme CLAE that leverages these examples for improved contrastive learning (CL).

## 3.1 Supervised learning

A classifier maps example $x \in \mathcal{X}$ into label $y \in \{1, \ldots, C\}$, where $C$ is a number of classes. A deep classifer is implemented by the combination of an embedding $f_\theta(x)$ of parameters $\theta$ and a softmax regression layer that predicts the posterior class probabilities

$$P_{Y|X}(i|x) = \frac{e^{w_i^T f_\theta(x)}}{\sum_k e^{w_k^T f_\theta(x)}}, \tag{1}$$

where $w_i$ is the vector of classification parameters of class $i$. Given a training set $\mathcal{D} = \{x_i, y_i\}$, the parameters $\theta$ and $\mathcal{W} = \{w_i\}_{i=1}^{C}$ are learned by minimizing the risk $\mathcal{R} = \sum_i L(x_i, y_i; \mathcal{W}, \theta)$ defined

by the cross-entropy loss

$$L_{ce}(x, y; \mathcal{W}, \theta) = -\log \frac{e^{w_y^T f_\theta(x)}}{\sum_k e^{w_k^T f_\theta(x)}}. \tag{2}$$

Given the learned classifier, the untargeted adversarial example $x^{adv}$ of a clean example $x$ is

$$x^{adv} = x + \delta \quad s.t. \quad ||\delta||_p < \epsilon \text{ and } \arg\max_i w_i^T f_\theta(x) \neq \arg\max_i w_i^T f_\theta(x_{adv}), \tag{3}$$

where $\delta$ is an adversarial perturbation of $L_p$ norm smaller than $\epsilon$. The optimal perturbation for $x$ is usually found by maximizing the cross-entropy loss, i.e.

$$\delta^* = \arg\max_\delta L_{ce}(x + \delta, y; \mathcal{W}, \theta) \quad s.t. \quad ||\delta||_p < \epsilon, \tag{4}$$

although different algorithms [79, 10] use different strategies to solve this optimization.

## 3.2 Contrastive learning

In SSL, the dataset is unlabeled, i.e. $\mathcal{U} = \{x_i\}$, and each example $x$ is mapped into an example pair $(x^p, x^q)$. In this work, we consider applications where $x$ is an image and the pair is generated by data augmentation. This consists of applying a transformation $p$ $(q)$ in some set of transformations $\mathcal{T}$ (e.g. spatial transformations, color transformations, etc.) to $x$, to produce the *augmentation* $x^p$ $(x^q)$. CL seeks to learn an invariant representation of image $x_i$ by minimizing the risk defined by the loss

$$L_{cl}(x_i^{p_i}, x_i^{q_i}; \theta, \mathcal{T}) = -\log \frac{\exp(f_\theta(x_i^{p_i})^T f_\theta(x_i^{q_i})/\tau)}{\sum_{k=1}^B \exp(f_\theta(x_k^{p_k})^T f_\theta(x_i^{q_i})/\tau)}, \quad p_i, q_i \sim \mathcal{T} \tag{5}$$

where $f$ is an embedding parameterized by $\theta$, $\tau$ is the temperature, $B$ is the batch size and $x_i^{p_i}, x_i^{q_i}$ are augmentations of $x_i$ under transformations $p_i, q_i$ randomly sampled from $\mathcal{T}$. Previous works on CL have considered many possibilities for the set of transformations $\mathcal{T}$. While [12] has shown that the choice of $\mathcal{T}$ has a critical role on SSL performance, most prior works do not give much consideration to the individual choice of $p_i$ and $q_i$, which are simply uniformly sampled over $\mathcal{T}$. In this work, we seek to go beyond this and select *optimal* transformations for each image $x_i$. More precisely, we seek augmentations that maximize the risk defined by the loss of (5), i.e.

$$\{p_i^*, q_i^*\} = \arg\max_{\{p_i, q_i\} \in \mathcal{T}} \sum_i L_{cl}(x_i^{p_i}, x_i^{q_i}; \theta, \mathcal{T}). \tag{6}$$

This is, in general, an ill-defined problem since, for each example $x_i$, what matters is the difference between the two transformations, not their absolute values.

## 3.3 Adversarial augmentation

This ambiguity can be eliminated by fixing one of the transformations of (5), i.e. solving instead

$$\{p_i^*\} = \arg\max_{\{p_i\} \in \mathcal{T}} \sum_i L_{cl}(x_i^{p_i}, x_i^{q_i}; \theta, \mathcal{T}), \quad q_i \sim \mathcal{T} \tag{7}$$

for a given set of $\{q_i\}$ sampled randomly from $\mathcal{T}$. However, it is usually difficult to search over the set $\mathcal{T}$ efficiently. Instead, we proposed to replace $x_i^{p_i}$ by an adversarial perturbation of $x_i^{q_i}$, i.e. find

$$x_i^{r_i} = \arg\max_{x \in \mathcal{A}(x_i^{q_i})} L_{cl}(x, x_i^{q_i}; \theta, \mathcal{T}), \quad q_i \sim \mathcal{T} \tag{8}$$

where $\mathcal{A}(x_i^{q_i})$ is a set of adversarial perturbations of $x_i^{q_i}$, defined by

$$\mathcal{A}(x) = \{x'|x' = x + \delta, ||\delta||_p < \epsilon\}. \tag{9}$$

In summary, given a set of transformations $\mathcal{T}$ and a set of augmentations $x_i^{q_i}$, the goal is to learn the adversaries $x_i^{r_i} = x_i^{q_i} + \delta_i^*$, by finding the perturbations $\delta_i^*$ that lead to the most diverse positive pairs $(x_i^{r_i}, x_i^{q_i})$. The rationale is that the use of these pairs in (5) increases the challenge of unsupervised learning, encouraging the learning algorithm to produce a more invariant representation.

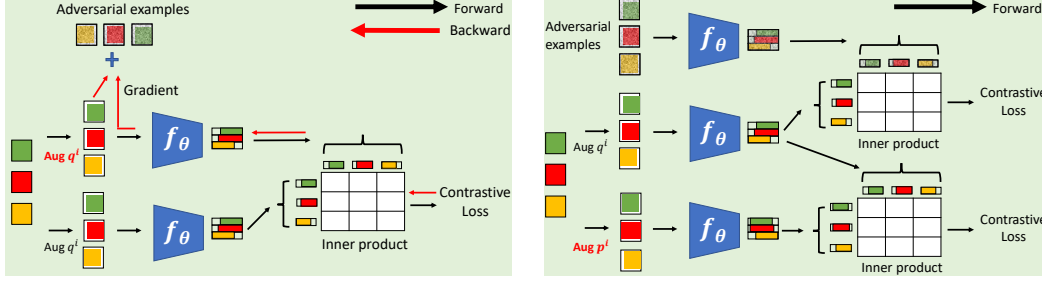

Figure 1: (Left) Generation of adversarial augmentations in step 4 of Algorithm 1 (Right) Adversarial training with contrastive loss in step 5 of Algorithm 1.

---

**Algorithm 1** Pseudocode of contrastive learning with adversarial example (CLAE) in a batch

---

1: **Input** $\mathcal{X} = \{x_i\}_{i=1}^B$, $AUG$:= Data Augmentation, Hyperparameter $\alpha$
2: $\mathcal{W}^p = \{x_i^{p_i}\}_{i=1}^B = AUG(\mathcal{X})$
3: $\mathcal{W}^q = \{x_i^{q_i}\}_{i=1}^B = AUG(\mathcal{X})$
4: Compute (15) with $\{x_i^{q_i}\}_{i=1}^B$ and $\mathcal{W}^q$ to obtain $\mathcal{W}^*$
5: Compute $L_{aug}$ of (16) with $\{x_i^{q_i}\}_{i=1}^B$ and $\mathcal{W}^p$, and $L_{adv}$ of (16) with $\{x_i^{q_i}\}_{i=1}^B$ and $\mathcal{W}^*$
6: Minimize (16) with hyperparameter $\alpha$

---

To optimize (8), we start by noting that the contrastive loss of (5) can be written as the cross-entropy loss of (2)

$$L_{cl}(x_i^{p_i}, x_i^{q_i}; \theta, \mathcal{T}) = -\log \frac{\exp(w_i^T f_\theta(x_i^{q_i}))}{\sum_{k=1}^B \exp(w_k^T f_\theta(x_i^{q_i}))} = L_{ce}(x_i^{q_i}, i; \mathcal{W}^p, \theta) \qquad (10)$$

by defining the classifier parameters $\mathcal{W}^p$ as $w_k = f_\theta(x_k^{p_k})/\tau$. As above, we can replace each $x_i^{p_i}$ by the optimal adversarial perturbation $x_i^{r_i} \in \mathcal{A}(x_i^{q_i})$, to obtain an optimal set of perturbed parameters $\mathcal{W}^* = \{f_\theta(x_k^{r_k})/\tau\}$ by solving

$$\{x_k^{r_k}\} = \underset{\{x_k \in \mathcal{A}(x_k^{q_k})\}}{\arg\max} \sum_i L_{ce}(x_i^{q_i}, i; \{f_\theta(x_k)/\tau\}, \theta), \quad q_i \sim \mathcal{T}. \qquad (11)$$

Note that this requires the determination of the optimal adversarial perturbation $x_i^{r_i}$ for the augmentation $x_i^{q_i}$ of each example $x_i$ in the batch. Using the definition of adversarial set of (9) in (11), results in the optimization

$$\{\delta_k^*\} = \underset{\{\delta_k\}}{\arg\max} \sum_i L_{ce}(x_i^{q_i}, i; \{f_\theta(x_k^{q_k} + \delta_k)/\tau\}, \theta) \quad s.t. \quad ||\delta_k||_p < \epsilon, \quad q_i \sim \mathcal{T}. \qquad (12)$$

This is an optimization similar to (4), but with a significant difference. While in (4) $\delta$ is a perturbation of the classifier input, in (12) it is a perturbation of the classifier weights. This implies that $\delta_k$ appears in the denominator of (10) for all $x_i^{q_i}$ in the batch and forces the optimization to account for all images simultaneously. In result, the embedding is more strongly encouraged to bring together the positive pairs $(x_i^{r_i}, x_i^{q_i})$ and separate all perturbations of different examples, i.e. the optimization seeks both challenging positive and negative pairs, performing hard negative mining as well.

### 3.4 FSGM attacks for unsupervised learning

The optimization of (12) can be performed for any set of transformations $\mathcal{T}$. Hence, the procedure can be applied to most CL methods in the literature. The optimization can also be implemented with most adversarial techniques in the literature. In this work, we rely on untargeted attacks with the popular fast gradient sign method (FGSM) [19]. For supervised learning, an untargeted FGSM attack consists of

$$x^{adv} = x + \delta = x + \epsilon \, sign(\nabla_x L_{ce}(x, y; \mathcal{W}, \theta)) \quad ||\delta||_2 < \epsilon, \qquad (13)$$

where $L_{ce}$ is the cross-entropy loss of (2). Similarly, to obtain $\delta_k^*$ of (12), the first order derivative of (12) is computed at $x_k^{q_k}$ to obtain $x_k^{r_k}$ with

$$x_k^{r_k} = x_k^{q_k} + \delta_k^* = x_k^{q_k} + \epsilon sign\left(\nabla_{x_k^{q_k}}\sum_i L_{ce}(x_i^{q_i}, i; \{f_\theta(x_k^{q_k})/\tau\}, \theta)\right) \quad (14)$$

$$= x_k^{q_k} + \epsilon sign\left(\nabla_{x_k^{q_k}}\sum_i L_{ce}(x_i^{q_i}, i; \mathcal{W}^q, \theta)\right), \quad ||\delta_k||_2 < \epsilon \quad (15)$$

Note that, due to this, the optimal set of adversarial augmentations $\mathcal{W}^* = \{f_\theta(x_k^{r_k})/\tau\}$ takes into consideration the relationship between all instances within the sampled batch.

### 3.5 Adversarial training with contrastive loss

To perform adversarial training, we adopt the training scheme of AdvProp [75], which uses two separate batch normalization (BN) layers for clean and adversarial examples. Unlike [75, 77], we set the momentum for the two BN layers differently. The momentum of the BN layer associated with clean examples is fixed to the value used by the original CL algorithm, while a larger momentum is used empirically for the BN layer associated with adversarial examples. The overall loss function is obtained by combining (10) and (15),

$$\arg\min_\theta \sum_{i=1}^B L_{ce}(x_i^{q_i}, i; \mathcal{W}^p, \theta) + \alpha \sum_{i=1}^B L_{ce}(x_i^{q_i}, i; \mathcal{W}^*, \theta) = \arg\min_\theta L_{aug} + \alpha L_{adv}, \quad (16)$$

where $\alpha$ balances between contrastive loss $L_{aug}$ parametrized with $\mathcal{W}^p$ and $L_{adv}$ parametrized with $\mathcal{W}^*$. The proposed procedure for CLAE is summarized in Algorithm 1 and visualized in Fig.1. Given augmentations $x_i^{q_i}$, adversarial examples are created by backpropagating the gradients of the contrastive loss to the network input. Contrastive loss terms are then computed for standard augmentation pairs and pairs composed by the augmentations $x_i^{q_i}$ and their adversarial examples, and combined with (16) to train the network.

### 3.6 Generalization to other contrastive learning approaches

While Algorithm 1 is based on the plain contrastive loss of (5), CLAE can be generalized to many other CL approaches in the literature. For example, UEL [78] uses an extra objective function to discourage different images from being recognized as the same instance. SimCLR [12] uses a projection head to avoid loss of information during pretext training and discards the projection head when optimizing the downstream task. To generalize Algorithm 1 to these approaches, it suffices to replace the plain contrastive loss with the loss functions on which they are based.

## 4 Experiments

In this section, we discuss an experimental evaluation of adversarial contrastive learning.

### 4.1 Setup

Experiments are performed on CIFAR10 [29], CIFAR100 [29] or tinyImagenet [1], using three different contrastive loss baselines: the loss of (5) (denoted as "Plain"), UEL [78] and SimCLR [12]. Unless otherwise noted, a Resnet18 encoder is trained using Algorithm 1 with $\alpha = 1$, standard Pytorch augmentation, and an adversarial batchnorm momentum of $0.01^1$. Two evaluation protocols are used, both based on a downstream classification task using features extracted by the learned encoder. These are implemented with a $k = 200$ nearest neighbor (kNN) classifier, and a logistic regression layer (LR). The encoder is trained with batch size 256 (128) and LR is trained for 1000 (200) epochs for CIFAR10 and CIRFAR100 (tinyImagenet). See supplementary for more details.

| Clean | Adv. | Diff. | Clean | Adv. | Diff. |
|-------|------|-------|-------|------|-------|

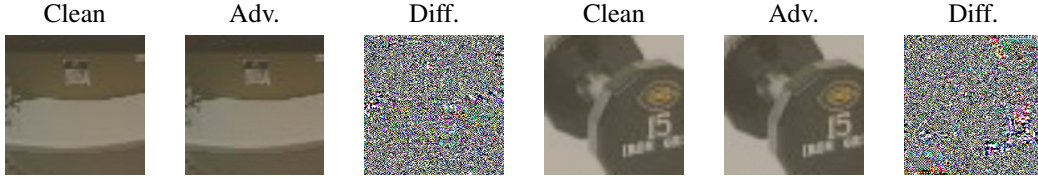

Figure 2: Adversarial examples computed with Algorithm 1 on tinyImagenet. The difference is amplified for visualization purpose.

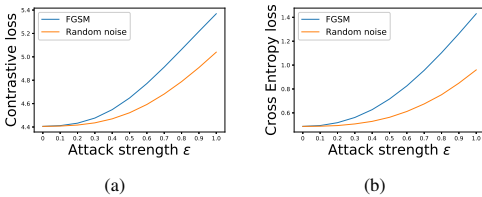

(a)

(b)

Figure 3: Effect of adversarial and random perturbations on (a) contrastive and (b) cross entropy losses.

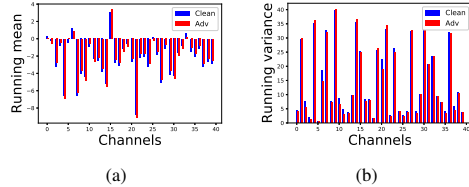

(a)

(b)

Figure 4: Running mean (a) and variance (b) of batch normalization for clean and adversarial examples, from randomly selected channels.

## 4.2 Influence of adversarial example

In this section, we study the effect of adversarial examples on both the SSL surrogate and downstream tasks. We compare the adversarial attacks of Algorithm 1 to random additive perturbations of the same magnitude. Fig. 3(a) shows the magnitude of the contrastive loss of the pretext task, on CIFAR10. For a given perturbation strength $\epsilon$, adversarial perturbations (blue) elicit a larger loss than random ones (red). Fig. 3(b) shows that, when the adversarial augmentations are fed into the downstream classification model, they elicit a larger cross entropy loss than random noise. This shows that adversarial augmentation produces more challenging pairs than random perturbations. Examples of adversarial augmentation are visualized in Fig. 2.

Nevertheless, there are some significant differences to the typical behaviour of adversarial examples in the supervised setting. First, the effect of adversarial attacks is weaker for SSL. In the SSL setting, adversarial examples only degrade the downstream classification accuracy by 20%. This is much weaker than previously reported for the supervised setting, using the adversarial examples of (4). Second, as illustrated in Fig. 4, the statistics of the batch normalization layers differ less between clean and adversarial examples than reported for supervised learning in [77]. Both of these observations are explained by the fact that, while the classifier parameters of (2) remain stable across training, those of (10) vary between batches. This creates uncertainty in the perturbation direction of (15), decreasing the differences between clean and adversarial attacks under the SSL setting.

## 4.3 Comparison to contrastive learning baselines

In this section, we investigate the gains of using CLAE framework for SSL, and the consistency of these gains across CL approaches. Table 1 shows the downstream classification accuracy for the two classifiers and three CL methods considered in these experiments. Note that adversarial training reduces to these methods when $\epsilon = 0$, in which case no adversarial examples are used. It can be seen that adversarial training improves the performance of all CL algorithms on all datasets, with a gain that is consistent across downstream classifiers. While best performance is achieved by using $\epsilon = 0.03$ in (15), $\epsilon = 0.07$ still beats the baseline in most cases.

## 4.4 Ablation study

An ablation study was conducted using the combination of SimCLR [12] and LR classifier. The study was performed on both CIFAR100 and tinyImagenet, with qualitatively identical results. We present CIFAR100 results here and tinyImagenet on the supplementary.

**Batch size** Fig. 5(a) shows that larger batch sizes improve the performance of both baseline and adversarial training. While this is consistent with the conclusions of previous studies on the importance of batch size [12, 26], adversarial training makes large batch sizes less critical. Besides outperforming the baseline for all batch sizes, it frequently achieves better performance with smaller sizes. For

Table 1: Downstream classification accuracy for three SSL methods, with and without ($\epsilon = 0$) adversarial augmentation, on different datasets.

| Method | $\epsilon$ | kNN | | LR | | |
| --- | --- | --- | --- | --- | --- | --- |
| | | Cifar10 | Cifar100 | Cifar10 | Cifar100 | tinyImageNet |
| Plain | 0 | 82.78±0.20 | 54.73±0.20 | 79.65±0.43 | 51.82±0.46 | 31.71±0.23 |
| | 0.03 | **83.09±0.19** | **55.28±0.12** | **79.94±0.28** | 52.04±0.32 | **32.82±0.10** |
| | 0.07 | 83.04±0.18 | 54.96±0.12 | 79.85±0.16 | **52.14±0.21** | 32.71±0.22 |
| UEL | 0 | 83.63±0.14 | 55.23±0.28 | 80.63±0.18 | 52.99±0.25 | 32.32±0.30 |
| | 0.03 | **84±0.15** | **55.96±0.06** | **80.94±0.13** | **54.27±0.40** | **33.72±0.30** |
| | 0.07 | 83.72±0.19 | 55.36±0.22 | 80.82±0.12 | 53.90±0.11 | 33.16±0.36 |
| SimCLR | 0 | 75.92±0.26 | 34.94±0.25 | 83.27±0.17 | 53.79±0.21 | 40.11±0.34 |
| | 0.03 | 76.45±0.32 | **38.89±0.25** | **83.32±0.26** | **55.52±0.30** | **41.62±0.20** |
| | 0.07 | **76.70±0.36** | 38.41±0.21 | 83.13±0.22 | 54.96±0.20 | 41.46±0.22 |

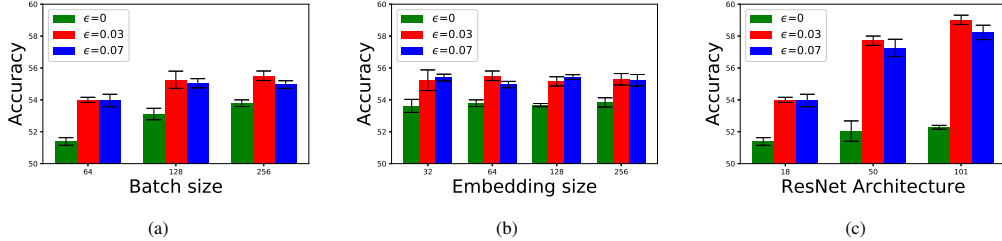

(a)  (b)  (c)

Figure 5: Ablation study of (a) batch sizes ,(b) embedding dimensions and (c) ResNet architectures.

example, adversarial training with $\epsilon = 0.03$ and batch size 64 (54%) outperforms the baseline with batch size 256 (53.79%). This suggests that adversarial training is both more robust and efficient.

**Embedding dimension** impact is evaluated in Fig. 5(b). This dimension does not seem to affect the performance of both baseline and adversarially trained model.

**Architecture** The impact of different architectures is evaluated in Fig. 5(c), showing that larger networks have better performance. However, the improvement is much more dramatic with adversarial training (red bar), where it can be as high as 5% (ResNet101 over ResNet18), than for the baseline (green bar), where it is at most 1%. This is likely because larger networks have higher learning capacity and can benefit from the more challenging examples produced by adversarial augmentation. Nevertheless, the fact that the ResNet18 with adversarial training beats the ResNet101 baseline shows that there is always a benefit to more challenging training pairs, even for smaller models.

**Hyperparameter** $\alpha$ weights the contributions of the two losses ($L_{aug}$ and $L_{adv}$) of (16). Fig. 6(a) shows downstream classification accuracy when $\alpha$ varies from 0 to 2, for $\epsilon = 0.03$. Accuracy starts to increase with $\alpha = 0.2$ and reaches its peak around $\alpha = 0.8$, but performance is fairly stable for $\alpha > 0.2$. There is no significant accuracy drop when the importance of adversarial examples is weighted by as much as twice that of clean examples ($\alpha = 2$). This shows that the features derived from the adversarial examples benefit learning.

**Attack strength** $\epsilon$ is studied in Fig. 6(b), which shows that adversarial training consistently beats the baseline ($\epsilon = 0$). Again, the performance is quite stable with $\epsilon$. This is possibly due to the affect of batch normalization, which aligns the statistics of embeddings from clean and adversarial examples.

**Epoch** While the default setting of SimCLR is to use 100 training epochs, there are usually gains in using longer pretext training, as shown in Fig. 6(c). While all methods benefit from this, adversarial training consistently beats the baseline. It is also observed that the larger perturbation of $\epsilon = 0.07$ benefits more from longer pretext training than smaller perturbations. Finally, training with adversarial

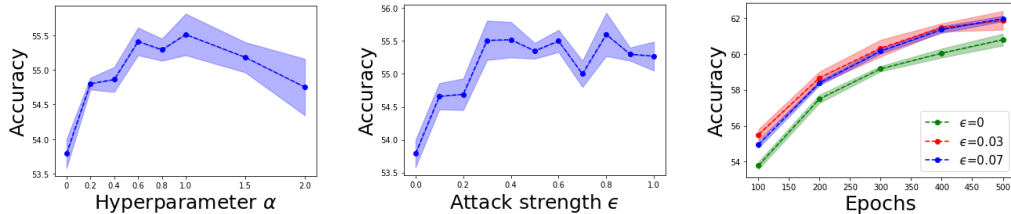

Figure 6: Ablation study for (a) hyperparameter $\alpha$, (b) attack strength $\epsilon$ and (c) longer pretext training.

Table 2: Comparison of transfer learning performance with linear evaluation to other image datasets.

| | CIFAR10 | CIFAR100 | Cars | Aircraft |
|---|---|---|---|---|
| SimCLR | 72.8±0.3 | 45.3±0.1 | 12.25 ±0.0 | 14.8±0.9 |
| CLAE | **73.6±0.3** | **47.0±0.2** | **12.60 ±0.1** | **16.2±0.6** |
| | DTD | Pets | Caltech-101 | Flowers |
| SimCLR | 50.6 ±1.2 | **44.4 ±0.4** | 68.2 ±0.3 | 46.0 ±0.3 |
| CLAE | **51.5±0.5** | **44.4±0.0** | **69.1±0.3** | **47.1±0.8** |

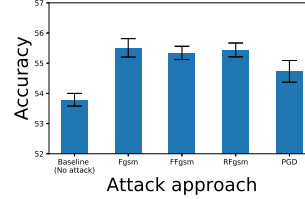

Figure 7: Comparison of different attack methods.

augmentation is more efficient. At 300 epochs, it achieves results close to those of the baseline with 400 epochs; with 400 epochs, it outperforms the baseline at 500 epochs. This is similar to previous observations in metric learning, where challenging training pairs are known to improve the both convergence speed and final embedding performance.

**Transfer to other downstream datasets** Transfer performance compares how encoders learned by different SSL approaches generalize to various downstream datasets. Following the linear evaluation protocol of [12], we consider the 8 datasets [29, 28, 38, 14, 49, 17, 46] shown in Table2. Both the encoder of SimCLR [12] and CLAE are trained on ImageNet100 (an ImageNet subset sampled by [63]), using a ResNet18. On ImageNet100, CLAE achieved 62.4±0.02 classification accuracy, outperforming SimCLR (61.7±0.02). This indicates that it can scale to large datasets. On the remaining datasets of Table2, it outperformed SimCLR on 7 out of the 8 datasets. This suggests that it generalizes better across downstream datasets. Since ImageNet100 does not contain any classes related to cars and airplanes, the performance on these 2 datasets is worse than on the others. In any case, CLAE beat [12] on several fine-grained datasets, such as Cars [28], Aircraft [38] and Flowers [46].

**Attack methods** While FGSM [19] is used in the above experiments, CLAE can be integrated with multiple attack approaches. To demonstrate this property, various attack methods (R-FGSM [65], F-FGSM [72], PGD [37]) were evaluated, with $\epsilon = 0.03$, on CIFAR100. As shown in Figure 7, R-FGSM and F-FGSM have performance comparable to FGSM, while PGD is slightly weaker. However, all these attack methods beat the baseline, indicating that the proposed framework learns a better representation.

# 5 Conclusion

In self-supervised learning (SSL), approaches based on contrastive learning (CL) do not necessarily optimize on hard negative pairs. In this work, we have proposed a new algorithm (CLAE) that generates more challenging positive and hard negative pairs, on-the-fly, by leveraging adversarial examples. Adversarial training with the proposed adversarial augmentations was demonstrated to improve performance of several CL baselines. We hope this work will inspire further research on the use of adversarial examples in SSL.

# 6 Broader Impact

This work advances the general use of deep learning technology, especially in the case that dataset annotations are difficult to obtain, and could have many applications. It advances several state of the art solutions on self-supervised learning (SSL), where no labels are provided. Moreover, while prior works in SSL suggest training with larger network, larger batch size and longer training epochs, the experiments in this works demonstrates that these factors are less critical by optimizing on effective training pairs. This can be beneficial in the scenario where time and gpu resource are limited. While this work mainly focuses on the study of image recognition, we hope this work can be extended to other application domains of SSL in the future.

# Acknowledgement

This work was partially funded by NSF awards IIS-1637941, IIS-1924937, and NVIDIA GPU donations. We also acknowledge and thank the use of the Nautilus platform for some of the experiments discussed above.

## Footnotes

[1]Or, equivalently 0.99 for Tensorflow implementation

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
