[Supplementary Material]

# Supplementary materials of
# Contrastive Learning with Adversarial Examples

**Chih-Hui Ho**    **Nuno Vasconcelos**
Department of Electrical and Computer Engineering
University of California, San Diego
{chh279, nvasconcelos}@ucsd.edu

## 1    Experiment details

All code are implemented with Pytorch [5] and ResNet18 [3] is used as default backbone unless specified. CIFAR10 [4], CIFAR100 [4] and tinyImagenet [1] are used for all contrastive learning baselines. Default $\alpha$ in Algorithm 1 is set to be 1. All codes are provided in the supplementary material.

To train the encoder with Algorithm 1, we implement the AdvProp[6] on ResNet architecture. For all batch norm layer in ResNet, an additional batch norm for adversarial example is created. While the batch norm (BN) momentum for clean example is the default Pytorch BN momentum (i.e. 0.1), the batch norm momentum for adversarial examples is set to be 0.01, indicating the update of running mean of adversarial BN is slower. Clean examples are forwarded to default BN and adversarial examples are forwarded to adversarial BN. Please refer to AdvProp[6] for more implementation details.

Three contrastive learning baselines are evaluated. **UEL** is the implementation of [7]. The public available code [1] is adopted. **Plain** is the implementation of (5). The hyperparameters and data augmentation detail is identical to [7]. **SimCLR** is the implementation of [2] by adopting the public available code[2].

For evaluation, k nearest neighbor (kNN) and minibatch logistic regression (LR) are considered. For kNN, the evaluation is identical to the protocol used in [7], while for LR, we train a single layer logistic regression with Adam optimizer on the embedding extracted from the fixed encoder, as in [2].

## 2    Quantitative results of Ablation study

The quantitative results of the ablation study conducted with SimCLR in the main paper are provided in section 2.1. In addition, the ablation study on tinyImagenet is provided in section 2.2. The experiment setting for each ablation study is described in detail.

| $\epsilon$ / Batch size | 64 | 128 | 256 |
|---|---|---|---|
| 0 | 51.39±0.24 | 53.11±0.36 | 53.79±0.211 |
| 0.03 | **54±0.16** | **55.26±0.54** | **55.51±0.30** |
| 0.07 | 53.96±0.39 | 55.04±0.29 | 54.96±0.20 |

Table 1: Quantitative results of Fig. 5 (a) on studying effect of different batch size on CIFAR100.

### 2.1    Ablation study on CIFAR100

Quantitative results of the ablation study on CIFAR100 in the main paper are shown below.

**Batch size** experiments are conducted on ResNet18 with embedding size 64. $\alpha = 1$ is used in Algorithm 1. Qualitative results are presented in Table 1.

**Embedding size** experiments are conducted on ResNet18 with batch size 256. $\alpha = 1$ is used in Algorithm 1. Qualitative results are presented in Table 2.

| $\epsilon$ / Embedding size | 32 | 64 | 128 | 256 |
|---|---|---|---|---|
| 0 | 53.62±0.41 | 53.79±0.21 | 53.67±0.10 | 53.84±0.29 |
| 0.03 | 55.23±0.65 | **55.51±0.30** | 55.16±0.29 | **55.29±0.36** |
| 0.07 | **55.4±0.21** | 54.96±0.20 | **55.43±0.15** | 55.23±0.36 |

Table 2: Quantitative results of Fig. 5 (b) on studying effect of different embedding size on CIFAR100.

**ResNet architecture** experiments are conducted with batch size 64 and embedding size 64. $\alpha = 1$ is used in Algorithm 1. Qualitative results are presented in Table 3.

| $\epsilon$ / Resnet | 18 | 50 | 101 |
|---|---|---|---|
| 0 | 51.39±0.24 | 52.04±0.64 | 52.28±0.12 |
| 0.03 | **54±0.16** | **57.71±0.29** | **59.02±0.29** |
| 0.07 | 53.96±0.39 | 57.26±0.54 | 58.23±0.45 |

Table 3: Quantitative results of Fig. 5 (c) on different architectures on CIFAR100.

**Hyperparameter** $\alpha$ experiments are conducted on ResNet18 with $\epsilon = 0.03$, batch size 256 and embedding size 64. Qualitative results are presented in Table 4.

| $\alpha$ | 0 | 0.2 | 0.4 | 0.6 |
|---|---|---|---|---|
| Accuracy | 53.79 ±0.21 | 54.80 ±0.08 | 54.86 ±0.18 | 55.41 ±0.21 |

| $\alpha$ | 0.8 | 1.0 | 1.5 | 2.0 |
|---|---|---|---|---|
| Accuracy | 55.29 ±0.16 | **55.51 ±0.30** | 55.18 ±0.22 | 54.75 ±0.40 |

Table 4: Quantitative results of Fig. 6 (a) on different hyperparameter $\alpha$ on CIFAR100 .

**Attack strength** experiments are conducted on ResNet18, batch size 256 and embedding size 64. $\alpha = 1$ is used in Algorithm 1. Qualitative results are presented in Table 5.

| Evaluation / $\epsilon$ | 0 | 0.01 | 0.02 | 0.03 | 0.04 | 0.05 | 0.06 | 0.07 | 0.08 | 0.09 | 0.1 |
|---|---|---|---|---|---|---|---|---|---|---|---|
| kNN | 34.94 | 38.89 | 38.35 | 38.89 | 38.99 | **39.03** | 38.65 | 38.41 | 38.76 | 38.47 | 38.57 |
| LR | 53.79 | 54.66 | 54.69 | 55.51 | 55.51 | 55.35 | 55.5 | 54.96 | **55.60** | 55.30 | 55.27 |

Table 5: Quantitative results of Fig. 6 (b) on different attack strength on CIFAR100.

## 2.2 Ablation study on tinyImagenet

In this section, the additional ablation study are conducted on tinyImagenet with SimCLR.

| $\epsilon$ / Batch size | 32 | 64 | 128 |
|---|---|---|---|
| 0 | 34.15±0.33 | 37.82±0.20 | 39.55±0.32 |
| 0.03 | 36.36±0.18 | 39.79±0.61 | **41.44±0.29** |
| 0.07 | **36.55±0.21** | **40±0.37** | 41.28±0.21 |

Table 6: Quantitative results on studying effect of different batch size on tinyImagenet.

**Batch size** experiments are conducted on ResNet18 with embedding size 64. $\alpha = 1$ is used in Algorithm 1. The proposed algorithm consistently beats the baseline ($\epsilon = 0$) about 2% across different batch sizes. Qualitative results are presented in Table 6.

**Embedding size** experiments are conducted on ResNet18 with batch size 128. $\alpha = 1$ is used in Algorithm 1. Similar to the observation in Cifar100 ablation study, different embedding size does not affect the trend. Qualitative results are presented in Table 7.

| $\epsilon$ / Embedding size | 32 | 64 | 128 | 256 |
|---|---|---|---|---|
| 0 | 39.54±0.28 | 39.55±0.32 | 39.79±0.14 | 39.46±0.27 |
| 0.03 | 41.15±0.21 | **41.44±0.29** | 41.49±0.11 | **41.37±0.19** |
| 0.07 | **41.21±0.25** | 41.28±0.21 | 41.62±0.26 | 40.72±0.46 |

Table 7: Quantitative results on studying effect of different embedding size on tinyImagenet.

**ResNet architecture** experiments are conducted with batch size 32 and embedding size 64. $\alpha = 1$ is used in Algorithm 1. While the gain is about 2.4% (36.55 vs 34.15) on ResNet18, the gain on ResNet50 increases to 3.9% (40.11 vs 44.01), which is similar to the observation in CIFAR100. Qualitative results are presented in Table 8.

| $\epsilon$ / Resnet | 18 | 50 |
|---|---|---|
| 0 | 34.15±0.33 | 40.11±0.64 |
| 0.03 | 36.36±0.18 | **44.01±0.31** |
| 0.07 | **36.55±0.22** | 43.68±0.24 |

Table 8: Quantitative results of the proposed method adopted to different architectures on tinyImagenet.

**Hyperparameter** $\alpha$ experiments are conducted on ResNet18 with $\epsilon = 0.03$, batch size 128 and embedding size 64. Again, the gain is stable for $\alpha > 0.2$. When $\alpha = 2$ is used, the contrastive learning still benefits from adversarial examples. Qualitative results are presented in Table 9.

| $\alpha$ | 0 | 0.2 | 0.4 | 0.6 |
|---|---|---|---|---|
| accuracy | 39.55 ±0.32 | 41.00 ±0.46 | 40.79 ±0.32 | 41.30 ±0.39 |
| $\alpha$ | 0.8 | 1.0 | 1.5 | 2.0 |
| accuracy | **41.69 ±0.09** | 41.44 ±0.29 | 41.39 ±0.02 | 41.08 ±0.25 |

Table 9: Quantitative results of the proposed method adopted to different hyperparameter $\alpha$ on tinyImagenet.

**Attack strength** experiments are conducted on ResNet18 with batch size 128 and embedding size 64. $\alpha = 1$ is used in Algorithm 1. Different $\epsilon$ has little influence on the final performance of the downstream task. Qualitative results are presented in Table 10.

| $\epsilon$ | 0 | 0.01 | 0.02 | 0.03 | 0.04 | 0.05 | 0.06 | 0.07 | 0.08 | 0.09 | 0.1 |
|---|---|---|---|---|---|---|---|---|---|---|---|
| LR | 39.55 | 41.17 | 41.42 | 41.44 | 41.41 | 41.45 | 41.25 | 41.28 | 41.37 | 41.26 | 41.34 |

Table 10: Quantitative results of the proposed method adopted to different attack strength on tinyImagenet.

**Downstream task training epoch** While the default training epoch of logistic regression for tinyImagenet on SimCLR is 200, longer training epoch (i.e 1000) is studied. While longer training epoch on downstream task boost all performances, the proposed method consistently beats the baseline, as shown in Table 11.

| Epoch /$\epsilon$ | 0 | 0.03 | 0.07 |
|---|---|---|---|
| 200 | 39.55±0.32 | **41.44±0.29** | 41.28±0.21 |
| 1000 | 40.11±0.34 | **41.62±0.20** | 41.46±0.22 |

Table 11: Quantitative results of the proposed method adopted to different attack strength on tinyImagenet.

## Footnotes

[1]https://github.com/mangye16/Unsupervised_Embedding_Learning

[2]https://github.com/Spijkervet/SimCLR