[Reviews · NeurIPS 2020]

Review 1

Summary and Contributions: This paper proposes using adversarial techniques to create positive and negative examples that are more difficult for a model being trained in a self-supervised fasion with contrastive loss to correctly classify. This, in effect, makes the SSL task more difficult and the authors demonstrate empirically that their methods leads to some modest accuracy gains on common image classification datasets. The main contribution is novel adaptation of adversarial techniques to perturb positive and negative example pairs such that they are more difficult to classify correctly.

Strengths: The strength of this paper is a convincing presentation of a novel idea with clear benefits as demonstrated by experiments. The authors show that this method can be applied to various different contrastive learning tasks and models. I think this paper would be interesting to the NeurIPS community.

Weaknesses: One weakness is that the authors do not evaluate their methods on larger models and datasets that most of the other SSL papers (such as SimCLR used as a baseline in this paper) commonly use. Therefore, it is difficult to say whether this approach scales to bigger model and data sizes.

Correctness: It appears, based on the authors evaluation, that the claims and the proposed method does work.

Clarity: The paper is well written, however it can be a relatively dense reading with a lot of notation, that (at least for me) was not the easiest read. Nonetheless I believe the authors managed to get their point across well. I would suggest adapting the main figure in the paper (Figure 1) to make it a bit more clear what's going on there and perhaps adjust the colors.

Relation to Prior Work: I found that this paper situates itself quite well in the literature, though I am less familiar with the adversarial learning literature.

Reproducibility: Yes

Additional Feedback: I've read the author rebuttal and thank the authors for their clarifications. I am believe my rating is still appropriate for this work.


Review 2

Summary and Contributions: This paper enhances various unsupervised learning methods with an additional adversarial data generation path and the associated loss. Contrastive learning is based on distinguishing different augmented versions of an image from other independently chosen images. This work proposes tweaking the augmented examples adversarially so that they are even more indistinguishable form other images (that are not augmented versions from the same image), that is "fooling" the classifier that distinguishes that augmented versions of the image from other images. The paper evaluates the quality of the pre-trained model by evaluating on the "downstream" supervised classification task

Strengths: This is clever method that works in several completely unsupervised setups with small, but consistent improvements across the board. The authors verify on downstream tasks for several datasets that the additional adversarial training improves the final classification score when the model is not fine-tuned. This is a strong indication that the method has true merits. The method is evaluated in multiple datasets and is used for augmenting multiple unsupervised approaches and exhibits the similar pattern: there is a sweet-spot for the size of the optimal adversarial perturbation and the improvements diminish if the perturbation exceeds that magnitude. The experiments are well designed and convincing and the method is novel.

Weaknesses: The paper shows compelling evidence that the adversarial training path helps the final downstream classification results when the model is not fine-tuned for supervised task. However any theoretical justification is absent. In general the idea is not extremely creative, but is certainly novel.

Correctness: The methodology has no obvious flaws and the results look consistent with rest of the literature. The numbers look good and give a clear supporting indication for the quality of the idea. However, no attempt at a mathematical analysis of the observed results or any explanation thereof is given.

Clarity: The paper is nicely written and motivated. Especially Figure 1 gives a very concise and intuitive description of the flow of the algorithm. The writing style is terse but easy to understand and focuses on the main features of the method. Overall, this paper is easy to follow and is convincing. In general, a pleasure to read.

Relation to Prior Work: The related works section is somewhat limited but it extensive enough to highlight the differences and similarities to previous methods. The method seems novel and interesting.

Reproducibility: Yes

Additional Feedback: The broader impact section misses the point, however I don't see any obvious ethical concerns regarding this research.


Review 3

Summary and Contributions: The papers improve contrastive learning by generating adversarial examples. It generates more challenging positive pairs and harder negative pairs. Experiments are conducted on three image classification dataset.

Strengths: 1. The ablation studies look sufficient. 2. The paper is well organized.

Weaknesses: [1] Adversarial Contrastive Estimation. ACL2018 [2] Self-supervised Approach for Adversarial Robustness CVPR 2020 1. Some related work. The paper states that " no attention has been previously devoted to the design of adversarial attacks for SSL". [2] has devoted to the design of adversarial attacks for SSL. Also, [1] "view contrastive learning as an abstraction of all such methods and augment the negative sampler into a mixture distribution containing an adversarially learned sampler. The resulting adaptive sampler finds harder negative examples, which forces the main model to learn a better representation of the data." This is not a problem considering the NIPS deadline. I am just write it down as an inference. It would be good if authors can explain the difference in later version. 2. Since the method insert adding adversarial noise into the data augmentation process of contrastive learning, a naive baseline of adding gaossian noise should be compared. Other noises should be considered too. 3. Since the improvement on cifar10 looks trivial, it would be great to see bigger improvement on more dataset. e.g. ImageNet, fine-graind dataset, etc. Is there a computation cost concern when apply your method to larger dataset? 4. Previous contrastive learning have results on transfer to a downstream task. ( ImageNet pretrain and detect on COCO ) Since you are compared with the naive contrastive learning method, it would be great if you can compare. 5. I understand that the computation cost of this method may be high. But since your major competitors are contrastive learning methods, it would be greater to compare with them under various dataset previous methods have used. SimCLR have results on (ImageNet Food CIFAR10 CIFAR100 Birdsnap SUN397 Cars Aircraft VOC2007 DTD Pets Caltech-101 Flowers). 6.If you can not show superior results on these datasets, why is this method useful, especially when your method is more complicated and take more computation time? 7. The experiments seem not very solid and promising to me. This is the major reasons I made my overall score. -------------After rebuttal----------------- I appreciate the author's rebuttal. It address some of my concerns. I would improve my rates to 4: An okay submission. Adding adversarial samples in SSL is not very novel. So I would expect a paper with solid experiments and extensive ablation studies. In rebuttal, authors conduct experiments on a small version of ImageNet due to hardware limitation. So whether it work on large scale dataset remain unknown. SSL is sensitive to parameters as shown in Fig.5. From the results on ImageNet100, I don't feel like there would be certain improvement on large scale dataset.

Correctness: yes

Clarity: yes

Relation to Prior Work: no

Reproducibility: Yes

Additional Feedback:


Review 4

Summary and Contributions: This paper introduce an adversarial attack mechanism to contrastive learning. By doing so, the model can have more challenging examples within a batch. The author also perform experiments on several datasets over baseline models.

Strengths: The paper is clear written. the proposed method is simple but reasonable. The experiments shows the improvement over baseline models.

Weaknesses: The choice of adversarial method FGSM needs further discussion, why this method is chosen, not other methods. The author may need to compare other baselines or statement, such as https://arxiv.org/pdf/1907.13625.pdf. Is it possible to also have adverbial examples on x_i?

Correctness: Yes

Clarity: yes

Relation to Prior Work: No, the author do not discuss much on other NCE enhanced algorithms.

Reproducibility: Yes

Additional Feedback:

[Author Response · NeurIPS 2020]

| | CIFAR10 | CIFAR100 | Cars | Aircraft |
|---|---|---|---|---|
| SimCLR | 72.8±0.3 | 45.3±0.1 | 12.25±0.0 | 14.8±0.9 |
| Ours | **73.6±0.3** | **47.0±0.2** | **12.60±0.1** | **16.2±0.6** |
| | DTD | Pets | Caltech-101 | Flowers |
| SimCLR | 50.6±1.2 | **44.4±0.4** | 68.2±0.3 | 46.0±0.3 |
| Ours | **51.5±0.5** | 44.4±0.0 | **69.1±0.3** | **47.1±0.8** |

Table 1: Comparison of transfer learning performance on 8 other datasets, using pretrained ResNet18 on ImageNet100.

Figure 1: Comparison of adversarial and Guassian perturbations.

Figure 2: Comparison of different attack methods.

We thank the reviewers for the very thoughtful comments. Major issues are addressed here; minor suggestions are
omitted (for space) and will be fixed as advised. **R1 Larger models**: For experiments with large ResNet models
(i.e. 50 and 101) see Fig 5(c) and L259-265. **R1&R3 Larger datasets**: SSL is very computing intensive in general.
SOTA results require very large datasets (ImageNet) and, more importantly, very large batch sizes (4,096). We are an
academic group. Although in a large university, our clusters are not large enough to run extensive experiments with
these settings, even for existing methods like SimCLR [12]. Nevertheless, to show that the proposed architecture works
for large datasets, we compared to SimCLR on ImageNet100 (an ImageNet subset sampled by [55]), using a ResNet18
under the linear evaluation. This achieved 62.4±0.02 classification accuracy, outperforming [12] (61.7±0.02), which
provides evidence that the proposed method scales up to big datasets. **R2 Theoretical justification**: We are working
on a theoretical analysis of the benefits of adversarial learning for SSL. However, this is not ready and would require
a paper of its own. We believe that an experimental showing of the benefits of adversarial examples is a first and
important preliminary step, which will be of interest to the NeurIPS audience and motivate others to work on the topic,
both experimentally and theoretically. **R3 & R4 Related work**: [2] was released after the NeurIPS deadline. More
importantly, it leverages SSL to increase robustness against adversarial attacks. This is a completely different goal form
our work (which leverages adversarial attacks to increase downstream task performance of SSL models). Ref [1] of
R3 is [7] in the submission. It uses a GAN-style manner for improving supervised learning problems in NLP, which
is again different from this work. [3] is an orthogonal work to ours. It studies the relationships between the infomax
criterion and minimization of the risk of (5), focusing on the impact of the choice of encoder and the tightness of mutual
information estimator (See section 3 of [3]). It concludes that infomax is insufficient for SSL. This is unrelated to our
work. It does highlight the importance of negative sampling (See section 4 and conclusion of [3]), which is one of
the appealing features of the proposed approach. We will cite and discuss these works. **R3 Guassian noise**: Good
suggestion! This was partially addressed in Fig. 3 of the submission (effect on loss of adversarial perturbations vs
uniform noise). To really compare the classification performance, we extended Fig 6(b) of the submission to show
the results obtained by adding guassian noise $\mathcal{N}(0, \epsilon)$ to the input image. As shown in Fig 1, this does not improve
classification performance. Instead, accuracy degrades as the perturbation magnitude increases. **R3 Computation**: The
proposed method requires an extra forward and backward pass per example during the pretraining stage. However,
this is not what prevents us from doing the large scale experiments. We can't do them even for standard SimCLR. We
note that pretraining cost is usually not seen as a major impediment in the literature, because the model is learned
once and can be transferred to many tasks. This is the reason why SimCLR levels of computation are tolerated, even
though few can afford to even perform the experiments at the scale needed to achieve SOTA results. **R3 Compare on
various datasets**: Good suggestion! These datasets are used to measure transfer performance. We followed the linear
evaluation protocol of [12], fixing the encoder pretrained on Imagenet100 and adding a linear layer, which is trained on
each downstream dataset. This was done for the encoders learned by both SimCLR and the proposed method, with
the results of Table1. The proposed approach outperformed [12] on 7 of the 8 datasets, indicating that the encoder
trained with the proposed method generalizes better across downstream datasets. Since ImageNet100 does not contain
any classes related to cars and airplanes, the performance on these 2 datasets is worse than on the others. In any case,
our results beat [12] on several fine-grained datasets, such as Cars, Aircraft and Flowers. **R3 superior results** As
shown above, we can show superior results for many SSL methods and downstream datasets. We cannot show SOTA
results because we lack Google scale resources, namely the ability to train on ImageNet with batch sizes of 4,096. **R4
Other attack method**: Good question. While all experiments in the submission use FGSM [17] for simplicity, many
untargeted attacks can be applied (See L182). This is now shown in Figure 2. Various attack methods (R-FGSM [57],
F-FGSM [4], PGD [1]) are compatible with the proposed framework, all beating the baseline. Various attack methods
lead to similar SSL performance. **R4 adversarial examples on $x_i$**: It is possible to compute the adversarial examples
on $x_i$, in Algorithm 1 (by forcing $x_i^{q_i} = x_i$). However, it is a common practice in SSL [12, 68, 20, 64, 34] (See L31) to
use one input example and one augmentation per pair. We simply follow this common practice.
46

[1] Aleksander Madry, Aleksandar Makelov, Ludwig Schmidt, Dimitris Tsipras, and Adrian Vladu. Towards deep learning models resistant to adversarial attacks. In
*International Conference on Learning Representations*, 2018.
[2] Muzammal Naseer, Salman Khan, Munawar Hayat, Fahad Shahbaz Khan, and Fatih Porikli. A self-supervised approach for adversarial robustness. In *IEEE/CVF
Conference on Computer Vision and Pattern Recognition (CVPR)*, June 2020.
[3] Michael Tschannen, Josip Djolonga, Paul K. Rubenstein, Sylvain Gelly, and Mario Lucic. On mutual information maximization for representation learning. *ArXiv*,
abs/1907.13625, 2020.
[4] Eric Wong, Leslie Rice, and J. Zico Kolter. Fast is better than free: Revisiting adversarial training. In *International Conference on Learning Representations*, 2020.


[Meta-Review · NeurIPS 2020]

It is an intuitive idea to combine adversarial learning with noise-contrastive learning. It would be an interesting contribution if this was not done before, and the reviewers mostly agree on this issue. While there have been concerns on the experimental setup being not mostly significant (regarding dataset size and model size), the reviewers tend to think the novelty of the proposed method outweighs the shortcomings.